# Femtosecond Time-Resolved Observation of Relaxation and Wave Packet Dynamics of the S1 State in Electronically Excited *o*-Fluoroaniline

**DOI:** 10.3390/molecules28041999

**Published:** 2023-02-20

**Authors:** Bumaliya Abulimiti, Huan An, Zhenfei Gu, Xulan Deng, Bing Zhang, Mei Xiang, Jie Wei

**Affiliations:** 1Xinjiang Key Laboratory for Luminescence Minerals and Optical Functional Materials, School of Physics and Electronic Engineering, Xinjiang Normal University, Urumqi 830054, China; 2School of Chemistry and Chemical Engineering, Xinjiang Normal University, Urumqi 830054, China; 3State Key Laboratory of Magnetic Resonance and Atomic and Molecular Physics, National Center for Magnetic Resonance in Wuhan, Innovation Academy for Precision Measurement Science and Technology, Chinese Academy of Sciences, Wuhan 430071, China; 4University of Chinese Academy of Sciences, Beijing 100049, China

**Keywords:** *o*-fluoraniline, relaxation dynamic, wave packet dynamic, quantum beat frequency, photoelectron image

## Abstract

Quantum beat frequency is the basis for understanding interference effects and vibrational wave packet dynamics and has important applications. Using femtosecond time-resolved mass spectrometry and femtosecond time-resolved photoelectron image combined with theoretical calculations, we study the electronic excited-state relaxation of *o*-fluoraniline molecule and the time-dependent evolution of vibrational wave packets between different eigenstates. After the molecule absorbs a photon of 288.3 nm and is excited to the S_1_ state, intramolecular vibrational redistribution first occurs on the time scale τ_1_ = 349 fs, and then the transition to the triplet state occurs through the intersystem crossing on the time scale τ_2_ = 583 ps, and finally, the triplet state occurs decays slowly through the time scale τ_3_ = 2074 ps. We find the intramolecular vibrational redistribution is caused by the 0^0^, 10b^1^ and 16a^1^ vibrational modes of the S_l_ state origin. That is, the 288.3 nm femtosecond laser excites the molecule to the S_1_ state, and the continuous flow of the vibrational wave packet prepares a coherent superposition state of three vibrational modes. Through extracting the oscillation of different peak intensities in the photoelectron spectrum, we observe reversible changes caused by mutual interference of the S_1_ 0^0^, S_1_ 10b^1^ and S_1_ 16a^1^ states when the wave packets flow. When the pump pulse is 280 nm, the beat frequency disappears completely. This is explained in terms of increases in the vibrational field density and characteristic period of oscillation, and statistical averaging makes the quantum effect smooth and indistinguishable. In addition, the Rydberg component of the S_1_ state is more clearly resolved by combining experiment and theory.

## 1. Introduction

Scientists have long hoped to observe the physical and chemical changes of molecules in real time and finally control these changes. In reality, observation and control are very difficult because the environments in which all physical and chemical changes occur are very complex, and the environment strongly influences chemical reactions. This complexity makes it difficult to understand the underlying mechanisms or how these reactions are related to each other. Even simpler physical and chemical changes do not occur in isolation. Wave packet dynamics is usually on the femtosecond (10^−15^ s; ultrafast) timescale, so conventional experimental methods do not yield ideal results. To study the wave packet dynamic of molecule, researchers have developed a series of femtosecond time-resolved (pump-probe) experimental methods such as femtosecond time-resolved photoelectron spectroscopy [1,2], femtosecond time-resolved photoelectron image (TRPEI) [3], and femtosecond time-resolved negative-ion velocity image [4,5]. When the bandwidth of the pump pulse contains multiple transition frequencies, the molecule is not excited to a quantum state corresponding to a single frequency but to a quantum state that is a superposition of such states. In a coherent superposition state, a wave packet is generated in which the position of the constructive interference changes over time, and its probe reveals an exponential decay in the intensity of the ionized signal with damped cosine oscillations with time [6,7]. This peculiar phenomenon of repeated oscillation is called quantum beat frequency and has attracted the attention of researchers in recent years [8,9,10].

*o*-Fluoroaniline is widely used in pharmaceuticals, pesticides, and dye intermediates. It has attracted attention from the scientific community because of their importance in industry, medicine and their special properties. For example, Lin’s team [11] determined the ionization energy of *o*-fluoroaniline molecule and vibrational energy level of cation by mass analysis threshold ionization spectrum and determined that their adiabatic ionization energy was 63,644 cm^−1^ (7.89 eV). Waware’s team [12] synthesized polyaniline-*o*-fluoroaniline by in-situ copolymerization of aniline and *o*-fluoroaniline through oxidation and found that the alternating current (AC) conductivity of the copolymer was greatly improved. Many studies have also been done on quantum beat frequency. Fengzi Ling’s team used femtosecond TRPEI to study the quantum beats of 2,4-difluorophenol [13], *o*-fluorophenol [14], and 2,4-difluoroaniline [9]. This research showed that beat frequency was caused by a change in molecular conformation. Shuai Li’s team used femtosecond TRPEI to study the wave packet evolution of pyrimidine [15], showing that beat frequency was caused by transitions between different vibrational energy levels of the ion state. The group of Junggil Kim studied the wave packet dynamic of 2-fluorothioanisole [16] via the femtosecond time-resolved pump-probe method and showed that there were two minimal energies when the molecule was in the S_1_ state. At these minima, the angles between the methyl group and the benzene ring were 0° and 51°, and the swinging motion of the methyl group made the molecule exhibit a beat frequency.

In summary, the studies on 2,4-difluorophenol, *o*-fluorophenol, 2,4-difluoroaniline, and 2-fluorothioanisole showed that quantum beat frequency could be directly observed through structural evolution, while the study of pyrimidine found that quantum beat frequency can also be observed through vibrational energy transitions between ion states. However, the beat signals all show a 180° phase difference in these studies, which can be associated with two structures or transition channels. On this basis, we pose two questions:
(1)Can quantum beat frequency be explained by the ionization cross-section?(2)Do in-phase signals appear in quantum beat phenomena?


The sensitivity of the photoelectron kinetic energy (PKE) distribution to vibrational dynamics has made femtosecond TRPEI very efficient and valuable for tracking excited state dynamics [17,18,19,20,21,22]. Because the 0^0^ vibrational mode of *o*-fluoroaniline has a higher ionization probability, while the 10b^1^ and 16a^1^ vibrational modes are inverse. When the molecule is pumped to the excited state, it may oscillate due to different ionization probabilities. The bandwidth of a femtosecond laser is so wide that it can perfectly include those vibration modes near the S_1_ state origin, forming a vibrational wave packet. In addition, the femtosecond laser reacts rapidly and can detect the evolution information of wave packets. On this basis, we use *o*-fluoroaniline as a model, conduct femtosecond TRPEI, and perform theoretical calculations to describe the relaxation dynamics of the molecular S_1_ state in detail.

## 2. Results and Discussion

### 2.1. Ground State Geometry of o-Fluoroaniline

Rotating the C-N bond of *o*-fluoroaniline in a 10° step to scan the total molecular energy with the density functional theory (DFT) and CAM-B3LYP/6-311+G(d,p) basis set, 36 structures are generated, as shown in Figure 1. It is specified that the two hydrogen atoms of the amino group are symmetrical on both sides of the benzene ring, and the angle is 0° at this time. Intuitively, there are two lowest-energy structures located at angles of 100° and 260°; *o*-fluoroaniline can form a double-well structure. When the angle of 100°, the energy is −10,524.46942 eV, and the energy is −10,524.46912 eV at a 260° angle. Thus far, we can accurately determine the ground state energy and structure of *o*-fluoroaniline. We find that the energy of the eleventh structure is the lowest, so this structure is the ground state geometry of *o*-fluoroaniline, and subsequent calculations are based on this structure.

In order to understand the excited state information of the molecule, we use time-dependent density functional theory (TD-DFT) to calculate the excitation energy (E), wavelength, oscillator strength (f) and other parameters of the first and second excited states of the molecule at the level of CAM-B3LYP/6-311G+(d,p) basis set as shown in Table 1. It can be seen that the vertical excitation energy of the S_1_ state is 4.25 eV, which is caused by the transition of orbital 29 → 30, with the transition wavelength center at 291.88 nm. And the vertical excitation energy of the S_2_ state is 5.09 eV, which is caused by the transition of orbital 29 to orbital 32, and the overall transition wavelength is 243.54 nm. Similarly, the vertical excitation energy of the T_1_ state is 3.45 eV, which is caused by the transition of orbital 30 to orbital 32, and the overall transition wavelength is 359.37 nm. We can find that the energy of the S_1_ state and the T_1_ state is relatively close, and the intersystem crossing (ISC) process may occur between them. Figure 2 shows the transition-related orbital shape. It should be noted that the conformations of the ground state, S_1_ state and T_1_ state calculated later have no imaginary frequency, indicating that our theoretical calculation is reasonable, as shown in Figure A1 in the Appendix A. Figure A2 in the Appendix A shows the molecular structure and Cartesian coordinate system of the ground state, S_1_ state, and T_1_ state.

### 2.2. Relaxation Kinetics of o-Fluoroaniline

Figure 3a shows the time decay curve of the parent ion under pump pulse at 288.3 nm and probe pulse at 800 nm. This line shape can be well fitted by convolution of a Gaussian cross-correlation function and three exponential decay functions, resulting in three-time constants, τ_1_ = 349 ± 17.43 fs, τ_2_ = 583 ± 29.16 ps, and τ_3_ = 2074 ± 103.69 ps. It is worth noting that τ_3_ decay for a long time, well above the upper limit of our delay platform, and therefore cannot be measured accurately. When the pump pulse is changed to 280 nm, the time decay curves of the parent ions are obviously different, as shown in Figure 3b. The Gaussian cross-correlation function and convolution of three exponential decay functions are also used to fit the data. The time scales of the three decay components are significantly different from the data of 288.3 nm, and their lifetimes are τ_1_ = 274 ± 13.71 fs, τ_2_ = 480 ± 24.00 ps, and τ_3_ = 3230 ± 161.48 ps. The τ_1_ and τ_2_ components at the pump pulse of 280 nm become faster, which is also found in previous work [9]. This is attributed to the differences in internal energy and vibrational density.

When the molecule is excited by two different pump pulses, τ_1_ is on the sub-picosecond scale, which is similar to the values for phenol [23] and derivatives [13,14,24,25,26,27], aniline [28] and derivatives [29,30], chlorobenzene [31]. There are also similarities in the structure between them, and predecessors attributed the sub-picosecond time scale to the process of intramolecular vibrational redistribution (IVR); therefore, we also attribute this to IVR. The theoretical data of Table 1 shows that the vertical excitation energy of the S_1_ state is 4.25 eV. The photon energy of 280 nm is 4.43 eV, and the 288.3 nm photon energy is 4.30 eV, which can only excite the S_1_ state of the molecule. Therefore, the τ_1_ component can be more precisely linked to the IVR of the wave packet from the vertical Franck-Condon (FC) region to the S_1_ origin of the lowest energy. There are two possible mechanisms for the τ_2_ component: one is an internal conversion to the S_0_ state. Given the very poor FC factor between the S_0_ state with high vibrational energy and the cationic state, it is difficult to ionize the thermal ground state molecules efficiently, which contradicts the existence of the τ_2_ component. The second possibility is ISC. It is worth noting that previously studied benzene [32], aniline [33], 2,4-difluoroaniline [29], and 2,5-difluoroaniline [27] and the time scale of 100 picoseconds was also obtained. Since the *o*-fluoroanilines that we studied are similar to their structures, and the time scale of hundred picoseconds is also obtained, we tend to the same attribution, ISC. The long life of the τ_3_ component is a direct consequence of the slow deactivation of the triplet state.

Figure 4 shows the photoelectron image when the pump pulse is 280 nm, the probe pulse is 800 nm, and the decay times are 0.027 ps and 1.429 ps. There are clearly seven spikes and one broadband in Figure 4a. The PKEs corresponding to the spikes are 0.18, 0.67, 0.98, 1.14, 1.67, 1.77, and 2.11 eV, and the broadband is located at 1.19~1.50 eV. We label these as the first to eighth peaks in ascending order of energy. It is worth noting that the PKEs of the first and seventh peaks differ by 1.59 eV, which is close to the 800 nm photon energy of 1.55 eV (the probe pulse is Gaussian). Thus, we believe that the first and seventh peaks correspond to the ionization of the same state. However, the first peak is derived from three photon probes, while the seventh peak originates from four photon probes. Therefore, we only need to discuss the first to the fourth, sixth, and eighth peaks. When we carefully observed Figure 4a,b, and found that the energies of the first, second, fourth to the sixth and eighth peaks decreased with the change in decay time, all of those peaks are associated with the S_1_ state. Moving on to the third peak, the energy remains basically unchanged, which is explained by the fact that energy from the triplet state flows into this peak. So, the third peak may be related to both the S_1_ state and the triplet state. The characteristics of these peaks also indicate that it is reasonable to attribute the τ_2_ component to ISC.

Each red arrow in Figure 4 represents the maximum PKE of the (1 + 3’) ionization process, which is calculated from:(1)$$E_{\mathrm{el}}^{\max}=hv_{288.3\mathrm{nm}}+3hv_{800\mathrm{nm}}-IP_{ad}$$
where $hv_{288.3\mathrm{nm}}$ is the energy of a pump photon, $hv_{800\mathrm{nm}}$ is the energy of a probe photon, and $IP_{ad}$ is the adiabatic ionization energy of 7.89 eV [12], which was determined from the mass-analyzed threshold ionization (MATI) spectrum. The first to fourth, sixth and eighth peaks are easily associated with the Rydberg states owing to their sharpness; that is, they have an accidental resonance with the Rydberg states. To accurately identify the corresponding Rydberg states, it is necessary to calculate the quantum defect δ via
(2)$$PKE=mhv_{800\mathrm{nm}}-R/{\left(n-\delta \right)}^{2}$$
where *m* is the number of probed photons, *R* is the Rydberg constant (13.606 eV), *n* is the principal quantum number, and *hν*_800 nm_ = 1.55 eV. The molecule can absorb two photons without being ionized; that is, *m* can be 2. For the first to fourth peaks, the suitable value of *n* is 3, and the calculated δ values are 0.84, 0.63, 0.47 and 0.36, respectively. About the sixth and eighth peaks, it is reasonable that the principal quantum number is only 4, that is *n* = 4, with resulting δ values of 0.91 and 0.29. In general, δ for second-group elements is 0.9~1.2 for s orbital, 0.3~0.5 for p orbital, and approximately 0 for d orbital [34]. Here, it is worth noting that the s, p, and d orbitals correspond to the principal quantum numbers of 1, 2, and 3, and the molecular orbitals are spherical, spindle-shaped, and plum-shaped. Therefore, the first to the fourth, sixth, and eighth peaks are identified as 3s, 3p_1_, 3p_2_, 3p_3_, 4s, and 4p, respectively. We also calculate the excitation energies of these peaks via:(3)$$T_{\mathrm{Rydberg}}=PKE+IP_{ad}-h\nu_{800nm}$$
where *T*_Rydberg_ is the excitation energy of the Rydberg state, and those are 4.97, 5.46, 5.77, 5.93, 6.46 and 6.90 eV, respectively.

Owing to the lack of experimental data, it is impossible to accurately judge the information of p and d orbitals. Fortunately, Tian Lu’s group [35] defined parameters such as the hole delocalization index (HDI) and electron delocalization index (EDI), which are very friendly for the theoretical detection of the Rydberg state. The smaller value of HDI (EDI) explains the higher level of hole (electron) delocalization, that is, the greater uniformity of distribution. Therefore, we should need large HDI and small EDI values so that the holes are very concentrated and the electrons are distributed in a certain space, which is the layout of the Rydberg state. We use DFT and CAM-B3LYP/aug-cc-pvtz [36] basis set to calculate some parameters of the HDI and EDI values and Rydberg state excitation energies, as shown in Table 2. The calculated energies of Rydberg states exhibit good agreement with the values obtained experimentally. It differs several times or even close to tenfold between HDI and EDI values, so we can preliminarily judge that these peaks are all in the Rydberg states, which is consistent with the experimental result. To distinguish the p and d orbitals, the molecular orbitals of the first, third, fourth, sixth and eighth peaks are analyzed and drawn with the software Multiwfn, as shown in Figure A3 in the Appendix A. We use the molecular orbital shape to assign the first, third, fourth, sixth, and eighth peaks to 3s, 3p_y_, 3p_x_, 4s, and 4p_y_. Therefore, the second peak is only 3p_z_, and the seventh peak is 3s.

We continue to change the pump pulse and measure the decay curve of the parent ion with time for a series of pump pulses, as shown in Figure 5. We find that the τ_3_ components at the pump pulses of 266 nm, 260 nm, 250 nm, and 240 nm are not much different from the previously measured pulses of 280 nm and 288.3 nm and are long-lived. While τ_1_ and τ_2_ are in the range of hundreds of picoseconds and a few seconds, they are gradually decreased by about the order of hundreds of femtoseconds. However, the S_1_ state is still excited. This is probably because the pump pulse is gradually shortened, thereby increasing the energy, so the molecular vibration energy increases and the decay time shorten. In addition, the τ_1_ component is getting shorter and shorter, which also shows that attributing τ_1_ to IVR is reasonable. Because a higher vibration will promote the occurrence of IVR, IVR will be faster.

### 2.3. Quantum Beat of the S_1_ State of o-Fluoroaniline

In the above, we have attributed the τ_1_ component to an IVR, and now we analyze this ultrafast formation in detail. We measure the photoelectron image at different time delays and extract the corresponding photoelectron energy spectrum, as shown in Figure 6a. This fig is integrated to obtain the signal intensity of the six peaks varying with delay time. After fitting with the cross-correlation function and the convolution of the three exponential decay functions (The three decay functions correspond to 349 fs, 583 ps, and 2074 ps), the decay components are subtracted to obtain the residual and the beat signals. In order to describe the beat phenomenon in detail, a fast Fourier transform (FFT) is performed on the extracted beat signal about all peaks (time domain signal is converted into frequency domain signal), as shown in Figure 6b. It is determined the frequency of the oscillating component in the frequency domain, and there are some frequencies that are 21, 52, 84, 105, and 126 cm^−1^, as shown in Table 3. Figure 6c is the result of five cosine function convolution cross-correlation functions, and the formula is given below:(4)$$I\left(\Delta t\right)=g\left(\Delta t\right)⊗(\overset{5}{\underset{i=1}{\sum }}A_{i}\cos\left(\left(2\pi v_{i}\Delta t\right)+\varphi_{i}\right)+C)$$
where *g*(∆*t*) is the pump-probe cross-correlation function, $v_{i}$ is the oscillation frequency of the *i* component, and $\varphi_{i}$ is the corresponding phase.

And we try to attribute the most likely origins of these frequencies. First, we locate the signal of 105 cm^−1^, which is compared with the MATI of the *o*-fluoroaniline [37]. The frequency observed is very close to the energy level difference (ΔE = 103 cm^−1^) between 0^0^ and 10b^1^ of the S_l_ state origin. About other frequencies, the group of Suzuki [38] studied the wavepacket dynamics of ^1^B_2_ (^1^Σ_u_^+^) of CS_2_ by sub-20 fs photoelectron imaging, also got some extra frequencies, they explained these by difference frequency and overtone frequency. The frequency of the 0^0^ vibration mode is 0, ν_1_ represents the 10b^1^ vibration mode, and the vibration frequency is 103 cm^−1^; ν_2_ represents the 16a^1^ vibration mode with a frequency of 246 cm^−1^. On the basis of this, we explain that 21, 52, 85, and 126 are $\frac{1}{2}$ × *ν*_2_ − *ν*_1_, $\frac{1}{2}$ × *ν*_1_, 2 × *ν*_1_ − $\frac{1}{2}$ × *ν*_2_ and $\frac{1}{2}$ × *ν*_2_, respectively. It should be noted that the true frequencies of 0^0^, 10b^1^, and 16a^1^ are 34,583, 34,686, and 34,829 cm^−1^ in the MATI spectrum, corresponding with energies of 4.29, 4.30, and 4.32 eV. Because the bandwidth of the femtosecond laser is relatively wide, when the pump pulse is 288.3 nm, 10b^1^ and 16a^1^ may be excited simultaneously. It can be seen that these frequencies are related to 10b^1^ and 16a^1,^ and our attribution of frequencies is reasonable. This indicates that the beat signal originates from the evolution of coherent vibrational wave packets by simultaneous excited the S_l_ 0^0^, S_l_ 10b^1^ and S_l_ 16a^1^ vibration modes.

Looking carefully at Figure 6c, we find that when the molecule is excited to the S_1_ state, first, it is in the FC region with a high vibrational state and relaxes to the more stable S_1_ state of the lowest energy. We also see that all the photoelectron peaks almost reach the maximum or minimum at the same time, which means that the beat signals of the peaks are in the same direction basically. A careful review of the MATI of the *o*-fluoroaniline cation [37] shows that the intensity of the S_l_ 0^0^ state is maximal, and the vibrational peaks of the S_l_ 10b^1^ and S_l_ 16a^1^ states are almost invisible in the spectrum. Therefore, the transition probability of the S_l_ 0^0^ state is much higher than that of the S_l_ 10b^1^ and S_l_ 16a^1^ states in *o*-fluoroaniline ionization, or the FC factor of the S_l_ 0^0^ state is larger [15], making the S_l_ 0^0^ state easier to ionize, and the S_l_ 10b^1^ and S_l_ 16a^1^ states are much less likely to ionize. When the wave packet is constantly flowing, there are two scenarios with delay: (1) when the energy in the wave packet is mainly concentrated near the S_l_ 0^0^ vibration mode, it can ionize the S_l_ 0^0^ vibration mode but cannot ionize the S_l_ 10b^1^ and S_l_ 16a^1^ modes, so the signal is at maximum, and (2) when the energy in the wave packet is mainly concentrated near the S_l_ 10b^1^ and S_l_ 16a^1^ vibration modes, the signal is at a minimum because neither of the three vibration modes can be ionized. The photoelectron signal continuously oscillates as the vibrational wave packets continuously flow. It is noted that the process is reversible because the wave packets are constantly flowing.

In order to determine the above beat frequency is the characteristics of the molecule itself, we change the pump pulse to 280 nm under the same experimental conditions. Figure 7a shows the time-resolved photoelectron energy spectrum, and the photoelectron signals (hollow circles) of six peaks varying with delay time are shown in Figure 7b. We also used three exponential decay functions to fit the data; the fitting results in Figure 7b are locked by three decay times of 274 fs, 480 ps, and 3230 ps. When the pump pulse is 280 nm, the quantum beat disappears because the energy of the photon with this pulse is sufficient to ionize the S_1_ 0^0^, S_1_ 10b^1^ and S_l_ 16a^1^ states.

There are two possible reasons for the disappearance of the quantum beat. First, the vibrational modes constituting the wave packet oscillate asynchronously, so the oscillation of the entire wave packet weakens or disappears. Second, as the vibrational state density increases, the statistical average of the quantum effect makes the femtosecond oscillation difficult to distinguish clearly at this time. The beat frequency is most obvious when the pump pulse is 288.3 nm. When the pump pulse is 280 nm, the beat frequency completely disappears. Therefore, the disappearance of the beat frequency may be due to the increase in the vibrational state density, and the statistical average begins to cover up the quantum effect. That increases the oscillation period to the point where the femtosecond oscillation can no longer be clearly distinguished, and the beat frequency is closely related to the pump pulse. In addition, from this work, we can find that the ionization efficiency of different vibration modes of molecules is different, which may also cause the phenomenon of beat frequency. Different molecules may have different properties, so the phase of the beat frequency may also be different. We will look for more detailed evidence in further work.

## 3. Experimental and Theoretical Methods

### 3.1. Experimental Parameters

The experiment was done on a self-made optoelectronic image device introduced in previous work [29,39]. This device was assembled by a supersonic molecular beam sampling system, vacuum system, ion lens, timing controller, 2-dimensional image probe system, signal acquisition system, and data processor. The vacuum chamber of the device has two parts: the beam source chamber and the ionization chamber, both of which are evacuated using a molecular pump (F200/1200) with a rotational speed of 400 r/s and a pumping speed of 1200 L/s. When no sample is injected, the vacuum level of both chambers is kept on the order of 10^−6^ Pa. When the pulse valve is opened to inject the sample, the vacuum level of the beam source chamber is between 1.0 × 10^−3^ Pa, and the ionization chamber is 1.0 × 10^−5^ Pa. The ionization chamber is shielded with a μ-metal (iron-nickel alloy) layer to protect the electrons from external electromagnetic fields.

The seed laser was generated by the self-mode-locked Ti: sapphire oscillator was amplified by a chirped-pulse regenerative amplifier, and the output was a fundamental frequency with a repetition frequency of 1 kHz, pulse width of 100 fs, a center wavelength of 800 nm, and single-pulse energy of 4.5 mJ/pulse. The fundamental laser consisted of two beams, one of which was used to pump the traveling-wave optical parametric amplifier system (TOPAS) to generate a pump pulse with a center wavelength of 288.3 nm, corresponding to the resonance excitation of the S_1_ state origin of the *o*-fluoroaniline molecule (34,583 cm^−1^). The other beam of the fundamental laser generated a probe pulse with a center wavelength of 800 nm. According to experience, the 800 nm probe pulse can cause some accidental resonances in the molecule. The polarization direction of the pump pulse and probe pulse was adjusted to be parallel to the probe pulse using a variable-wave plate and a half-wave plate, respectively. The photoelectron image was collected under different pump-probe time delays, and then each three-dimensional image was reconstructed with a basis set expansion (BASEX) transformation [40]. The typical cross-correlation is measured to be 152 ± 25 fs obtained by non-resonant ionization of Xe.

*o*-Fluoroaniline, with a purity of 99.9%, was used as a sample without further purification. The background pressure was 2 atm (1 atm = 101,325 Pa), and helium was used as the carrier gas. The saturated vapor of the mixture of the liquid sample and carrier gas was injected into the beam source chamber through a pulse valve to form a supersonic molecular beam. The supersonic molecular beam then passed through a skimmer with a radius of 1 mm. This skimmer was collimated into the ionization chamber between the ion lens repelling pole (R) and accelerating pole (E), where it interacted with the laser beam. The *o*-fluoroaniline molecule was ionized under the pump pulse and the probe pulse to generate photoelectrons and photoions, and the ion lens accelerated and focused the photoelectrons and ions on the two microchannel plates (MCPs). A high-resolution photoelectron image or ion image was formed on a two-dimensional position-sensitive probe consisting of a fast-response P47 phosphor screen. The voltages of the three plates of the ion lens during the collection of photoelectron images were V_G_ = 0 V, V_E_ = −2758 V, and V_R_ = −4000 V. The photoelectron image was collected by a CCD camera mounted on the back of the probe, and a photomultiplier tube was also used to collect photoelectron/ion mass spectrometry signals. The sequence of the whole system was controlled by a DG535 controller.

### 3.2. Theoretical Calculation

To find the lowest-energy structure of the ground state of *o*-fluoroaniline, we use Gaussian 09 [41] and Molclus 1.9.9.7 [42] to perform a conformation search on the molecule by the DFT and CAM-B3LYP/6-311+G(d,p) [43,44,45] basis set. We also calculate some parameters of the S_1_ and S_2_ states based on TD-DFT and CAM-B3LYP/6-311G+(d,p) basis set. To determine the molecular p and d orbital information, we optimize and calculate the excitation energy of the Rydberg state based on the founding structure and DFT CAM-B3LYP/aug-cc-pvtz basis set. The wave function information of the molecules of the Rydberg state is imported into the Multiwfn 3.7 [35,46] program package, and these orbitals are analyzed and plotted.

## 4. Conclusions

Studying molecular excitation state dynamics can help in understanding the mechanisms of photophysical and photochemical reactions at the molecular level, thus is possible to regulate chemical reactions. This is significant to life science, material science, and environmental science. Focusing on relaxation dynamics, wave packet evolution, and other issues, we use femtosecond time-resolved mass spectrometry and femtosecond TRPEI combined with theoretical calculations to study the excited state of the *o*-fluoroaniline molecule. In the experiment, a 288.3 nm femtosecond pulse is used to excite the molecule to the S_1_ state, IVR first occurs on the timescale of 349 fs, after which it transits to the triplet state through ISC on the timescale of 583 ps, and the triplet decays slowly through a timescale of 2074 ps. The IVR is caused by the 0^0^, 10b^1^ and 16a^1^ vibrational states of the S_l_ state origin; that is, the 288.3 nm femtosecond laser excites the molecule to the S_1_ state and prepares a coherent superposition state. Reversible changes are observed through the oscillation of different photoelectron peaks in the photoelectron energy spectrum. When the pump pulse is 280 nm, the 0^0^, 10b^1^, and 16a^1^ vibrational modes are ionized, and the beat frequency completely disappears. In this experiment, we found the wave packet dynamics process formed by three vibration modes different from the previous ones. And experiment and theory are combined for the first time to more clearly resolve some Rydberg states of the S_1_ state. This work provides an important reference for interpreting the excited-state dynamic of other fluorine-containing aniline molecules and resolving the Rydberg state.

## Figures and Tables

**Figure 1 molecules-28-01999-f001:**
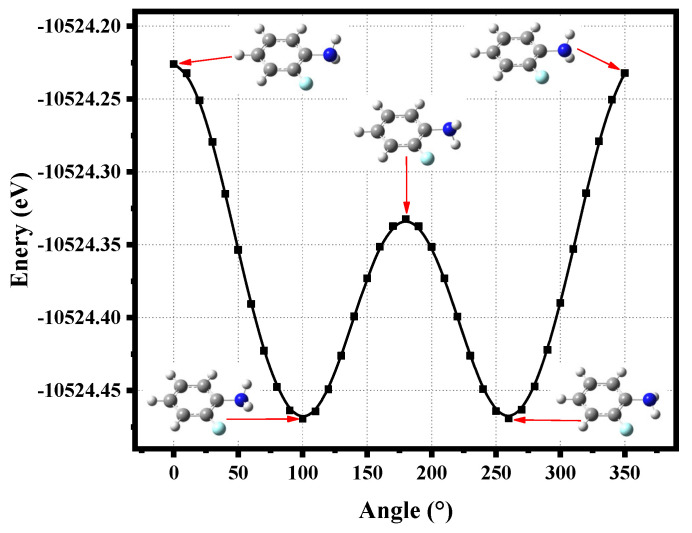
*o*-Fluoroaniline energy varies with angle, and the five structures of the molecule are given. Calculation method and basis set: DFT and CAM-B3LYP/6-311+G(d,p).

**Figure 2 molecules-28-01999-f002:**
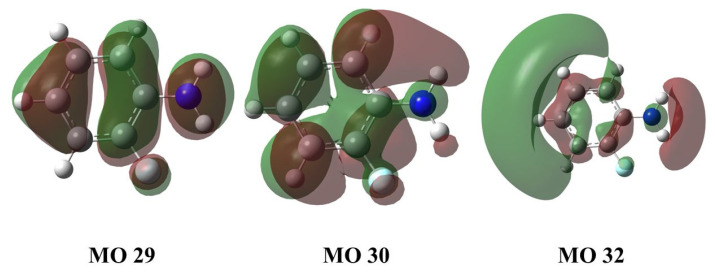
MO29, MO30, MO32 orbitals of *o*-fluoroaniline S_1_ state. Calculation method and basis set: TD-DFT and CAM-B3LYP/6-311+G(d,p).

**Figure 3 molecules-28-01999-f003:**
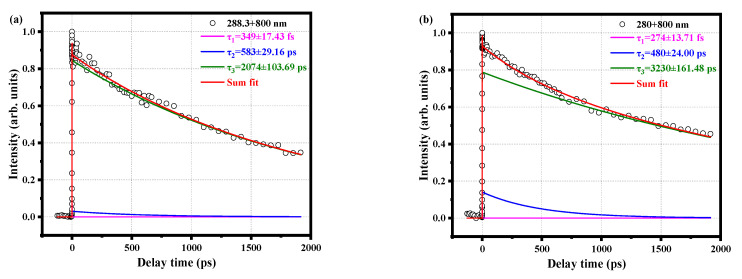
Time decay curves of the parent ion measured with (**a**) a 288.3 nm pump pulse and 800 nm probe pulse, (**b**) a 280 nm pump pulse and 800 nm probe pulse. The circles are experimental data, and the solid lines are fitted data. Both curves are fitted using a Gaussian cross-correlation function and a convolution of three exponential decay functions, resulting in different decay times.

**Figure 4 molecules-28-01999-f004:**
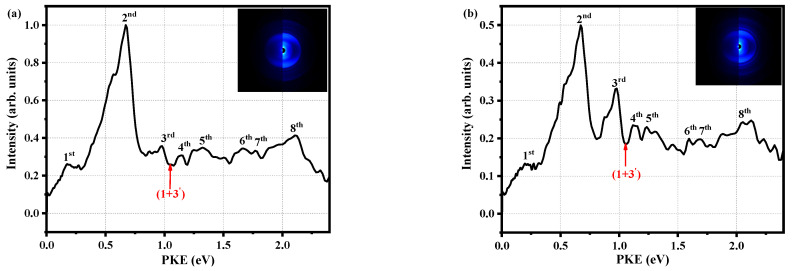
Photoelectron spectra at delay times of (**a**) Δt = 0.027 ps and (**b**) Δt = 1.429 ps. The ordinates of the two plots are normalized, and (**b**) is normalized with the maximum value of (**a**) as a reference. The insets are the corresponding optoelectronic images (the original image on the left and BASEX transformed image on the right).

**Figure 5 molecules-28-01999-f005:**
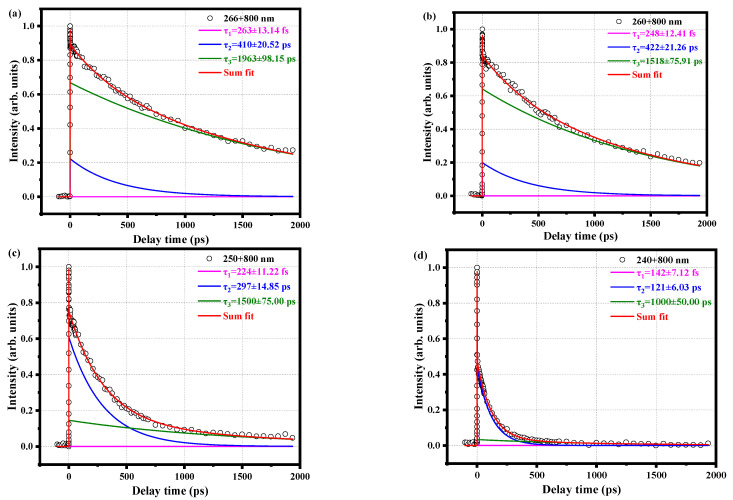
Time decay curves of the parent ion measured with (**a**) 266 nm pump pulse and 800 nm probe pulse, (**b**) 260 nm pump pulse and 800 nm probe pulse, (**c**) 250 nm pump pulse and 800 nm probe pulse, and (**d**) 240 nm pump pulse and 800 nm probe pulse. The circles are experimental data, and the solid lines are fitted data. All curves are fitted using a Gaussian cross-correlation function and a convolution of three exponential decay functions to obtain different decay times.

**Figure 6 molecules-28-01999-f006:**
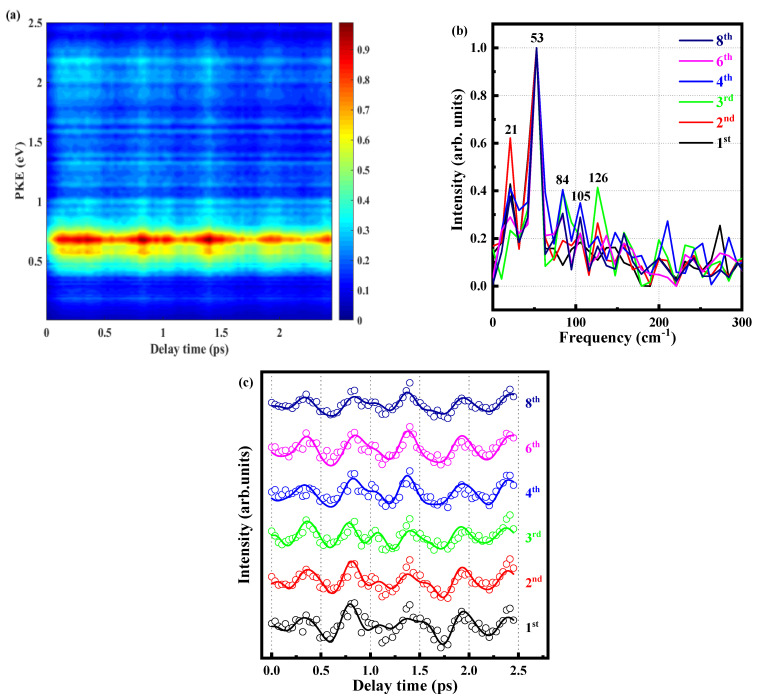
(**a**) Time-resolved photoelectron spectrum obtained with a 288.3 nm pump pulse and 800 nm probe pulse. (**b**) The spectrum obtained by Fourier transform of the residual data at the fourth peak. (**c**) Signal intensities of the six peaks as functions of delay time. Experimental data are given as open circles, while solid lines are fitted results.

**Figure 7 molecules-28-01999-f007:**
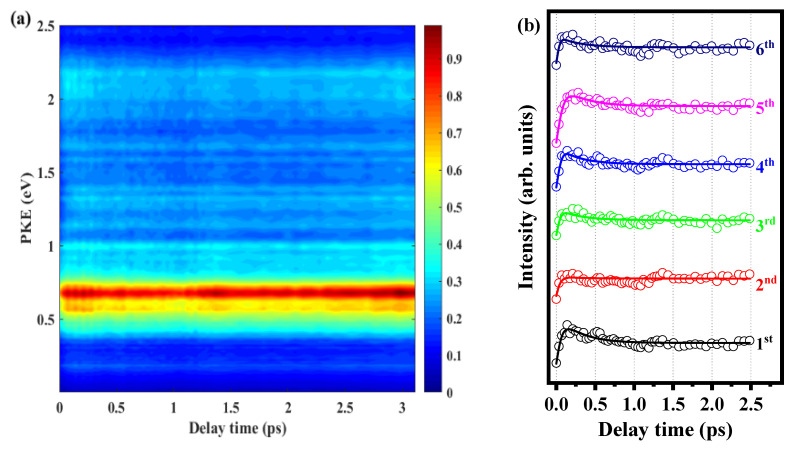
(**a**) Time-resolved photoelectron spectrum obtained with 280 nm pump pulse and 800 nm probe pulse. (**b**) Signal intensities of the six peaks as functions of delay time. Experimental data are given as open circles, while solid lines are fitted results.

**Table 1 molecules-28-01999-t001:** Wavelength, vertical excitation energy (E), oscillator strength (f) and transition orbital of S_1_, S_2_ and T_1_ states of *o*-fluoroaniline molecules. Calculation method and basis set: TD-DFT and CAM-B3LYP/6-311+G(d,p).

State	Transition	Wavelength/nm	E/eV	f
S_1_	29 → 30	291.88	4.25	0.0600
S_2_	29 → 32	243.54	5.09	0.0107
T_1_	30 → 32	359.37	3.45	0.0023

**Table 2 molecules-28-01999-t002:** Theoretically calculated molecular hole delocalization index (HDI), electron delocalization index (EDI), and Rydberg energy (E), and experimentally obtained Rydberg excitation energy (T). The E represents the theory calculated value, and T is the Rydberg excitation energy calculated according to Equation (3) for the experimental data. The theoretical method is used the basis set DFT and CAM-B3LYP/aug-cc-pvtz.

		Theoretical Value			Experimental Value			
Peaks	PKE (eV)	HDI	EDI	E (eV)	*n*	δ	T (eV)	States
1st	0.18	11.31	2.72	5.11	3	0.84	4.97	3s
2nd	0.67				3	0.63	5.46	3p_z_
3rd	0.98	9.64	3.84	5.70	3	0.47	5.77	3p_y_
4th	1.14	10.31	1.79	5.88	3	0.36	5.93	3p_x_
6th	1.67	8.20	1.61	6.44	4	0.91	6.46	4s
7th	1.77	11.31	2.72	5.11	3	0.83	5.01	3s
8th	2.11	8.72	2.99	6.67	4	0.29	6.90	4p_y_

**Table 3 molecules-28-01999-t003:** The frequency of FFT and their allocation. ν_1_ represents 10b^1^ vibration mode, and the vibration frequency is 103 cm^−1^; ν_2_ represents the 16a^1^ vibration mode with a vibration frequency of 246 cm^−1^.

Vibrational State	$\frac{1}{2}\times v_{2}-v_{1}$	$\frac{1}{2}\times v_{1}$	$2\;\times \;v_{1}-\frac{1}{2}\times v_{2}$	*v* _1_	$\frac{1}{2}\times v_{2}$
Frequency (cm^−1^)	21	52	84	105	126
Periodicity (fs)	1588	642	397	318	265

## Data Availability

Data sharing not applicable.

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
