# Peer review of "Femtosecond Time-Resolved Observation of Relaxation and Wave Packet Dynamics of the S1 State in Electronically Excited o-Fluoroaniline"

_molecules, 2023, doi:10.3390/molecules28041999_

Round 1

Reviewer 1 Report

The manuscript "Femtosecond time-resolved observation of relaxation and wave packet dynamics of the S1 state in electronically excited o-fluoroaniline" by Abulimiti et al. presents an joint experimental/theoretical investigation of the electronic excited-state relaxation of o-fluoroaniline.

For the experimental part the Authors present fentosecond time-resolved mass spectrometry and photoelectron image measurements.

For the theoretical part, the Author base their arguments on (TD-)DFT calculations.

Although the experiments show some good insights, the language is very poor and the theoretical part has too many flaws. Therefore, I can only recommend its acceptance after major revisions.

A few points to be considered to improve their manuscript, in addition to an extensive revision of the English language, are the following:

a) The authors write "DFT CAM-B3LYP/6-311G+(d,p) basis set", but the references for the used functional and basis sets employed are not cited. The Author only cite the computational program they use.

b) For the minimum geometry, did they performed a relaxed or a rigid scan on the rotated bond? Are all coordinates optimized for a given rotated angle or they just compute the energy for a given conformation? How does the minimum energy of their scan compares with the energy on the relaxed geometry assuming no symmetry point group?

c) For the calculation of excited-states, the authors describe the character of the excitation in terms of "orbits". The appropriate term is orbital. Moreover, they write transition 29->30 or 29->32 but they do not show/describe how these orbitals 29, 30, and 32 looks like.

d) Using TD-DFT the authors can estimate the gap between the S1/T1, where T1 is the triplet state reached by intersystem crossing using either the ground state equilibrium geometry of at the equilibrium geometry of the S1 state. Then, describe the possibility of intersystem crossing based in these energy gaps.

e) Describe how the HDI and EDI parameters shown in Table 2 are computed.

 f) In lines 241 to 243, please, elaborate the association of the peaks with Rydberg structures. What is the meaning of "accidental resonance with Rydberg states"?

Reviewer 2 Report

I was asked to provide a review for the manuscript ID molecules-2191003 by Abulimiti et al. entitled “Femtosecond time-resolved observation of relaxation and wave packet dynamics of the S1 state in electronically excited o-fluoroaniline” In this contribution, authors studied the electronic excited-state relaxation of o-fluoroaniline molecules and the time-dependent evolution of vibrational wave packets between different eigenstates using femtosecond time-resolved mass spectrometry and femtosecond time-resolved photoelectron image combined with theoretical calculations. Good scientific effort has been made by experimental and theoretical scientists. This research article is properly handled by the authors. Results are accurate. To sum up, I believe that this article fulfills the high-quality standards of Molecule and can be of interest for its readership.

Recommendation: I recommend this paper for publishing in Molecule with following minor improvements.

1.     Provide frequency analysis graphs showing absence of imaginary frequencies in supplementary Information file.

2.     It is advised to provide the Cartesian coordinates of all investigated systems in supplementary Information file.

3.     Abstract and Conclusion sections are weak. It is advised to incorporate the vital findings.

4.     The manuscript clearly would benefit from a careful read through and edit because It contains grammatical and typographical errors.

5.     Please indicate the full name of each abbreviation at its first appearance in the draft too.

6.     All Figures and Tables in the manuscript are needed to magnify and properly arrange to make it suitable for reading.

7.     Equations size are outsized.

8.     The authors should use all symbols in italic format throughout the manuscript.

9.      References should be written in accordance with the requirements of the journal.

Round 2

Reviewer 1 Report

The current version of the manuscript has improved after the authors addressed the suggestions of both reviewers.

However, as also suggested by the second reviewer, the English could be improved, and I do not think it has been improved since the first version.

I would like to give the authors a chance to read the manuscript again.

To give a few examples, in the abstract, "finally the triplet state occurs decays slowly" reads better as "finally the triplet state decays slowly".

"Through extract the oscillation of different peaks intensities in the photoelectron spectrum" could be replaced with "By extracting the intensities' oscillations of different peaks in the photoelectron spectrum"

Moreover, the sentence in line 84 is difficult to understand.

In line 239 replace "we think" with "we believe".

In lines 286-287, "The Rydberg state energies of theoretically calculation are agreed well with the values by experimentally obtained" can be rewritten as "The calculated energies of Rydberg states exhibit good agreement with the values obtained experimentally".

And so on.

After a revision of the English language, I would be happy to warrant the acceptance of the manuscript.